# Cloning and Characterization of Cellulase from *Paenibacillus peoriae* MK1 Isolated from Soil

**Sang Jin Kim** [1,†]**, Kyung-Chul Shin** [2,†]🆔**, Dae Wook Kim** [3]**, Yeong-Su Kim** [3,*]🆔 **and Chang-Su Park** [1,4,*]

[1] Department of Food Science and Technology, Daegu Catholic University, Gyeongsan 38430, Republic of Korea; ksangjin92@naver.com

[2] Department of Integrative Bioscience and Biotechnology, Konkuk University, Seoul 05029, Republic of Korea; hidex2@naver.com

[3] Industrialization Research Division, Baekdudaegan National Arboretum, Bonghwa 36209, Republic of Korea; dwking@koagi.or.kr

[4] Department of Pharmaceutical Science and Technology, Daegu Catholic University, Gyeongsan 38430, Republic of Korea

* Correspondence: yskim@koagi.or.kr (Y.-S.K.); parkcs@cu.ac.kr (C.-S.P.)

† These authors contributed equally to this work.

**Abstract:** An isolated bacterium from soil that highly hydrolyzes cellulose was identified as *Paenibacillus peoriae* and named *P. peoriae* MK1. The cellulase from *P. peoriae* MK1 was cloned and expressed in *Escherichia coli*. The purified recombinant cellulase, a soluble protein with 13.2-fold purification and 19% final yield, displayed a specific activity of 77 U/mg for CM-cellulose and existed as a metal-independent monomer of 65 kDa. The enzyme exhibited maximum activity at pH 5.0 and 40 °C with a half-life of 9.5 h in the presence of $Ca^{2+}$ ion. The highest activity was observed toward CM-cellulose as an amorphous substrate, followed by swollen cellulose, and sigmacell cellulose and α-cellulose as crystalline substrates. The enzyme and substrate concentrations for the hydrolysis of CM-cellulose were optimized to 133 U/mL and 20 g/L CM-cellulose, respectively. Under these conditions, CM-cellulose was hydrolyzed to reducing sugars composed mostly of oligosaccharides by cellulase from *P. peoriae* MK1 as an endo-type cellulase with a productivity of 11.1 g/L/h for 10 min. Our findings will contribute to the industrial usability of cellulase and the research for securing cellulase sources.

**Keywords:** cellulase; *Paenibacillus peoriae*; characterization; CM-cellulose





## 1. Introduction

Cellulose, which is a polysaccharide consisting of glucose linked by β-1,4 glycosidic linkages, is the most abundant plant biomass on earth that will not be depleted as long as the sun exists. Cellulose, a renewable and usable natural resource, has attracted considerable attention because of its potential as an alternative fuel and as a raw material for the industrial production of bio-based functional materials. [1,2]. For industrial use, cellulose must be decomposed into monosaccharides or low-molecular-weight structures. However, the degradation of cellulose is not straightforward because it is a polysaccharide with a high degree of polymerization caused by glycosidic bonds ranging from 250 to 1000; polysaccharides are connected to each other by hydrogen bonds. In particular, there are many problems associated with the industrial use of cellulose because of the difficulty in decomposing cellulose [3–5]. For this reason, research on cellulose degradation by chemical methods such as heating and acid and alkali treatments has been continuously conducted to render the cellulose useful. However, with industries becoming more environment-friendly and safer working environments, biological degradation methods using cellulases have garnered considerable attention.

Cellulases constitute a broad family of enzymes belonging to the O-glycoside oxidases (EC 3.2.1) and have diverse members classified into several types including endoglucanase

(1,4-β-D-glucan-4-glucanohydrolase; CMCase; EC 3.2.1.4), exoglucanase (1,4-β-D-glucan glucohydrolase; cellobiohydrolase; EC 3.2.1.74), and β-glucosidase (β-D-glucoside gluco-hydrolase; EC 3.2.1.21) according to the type of reaction they catalyze. In many bacteria, the cellulase complex, composed of different enzymatic subunits, acts cooperatively on cellulose to break it down into D-glucose [5]. Based on these characteristics, cellulase has been continuously used as an eco-friendly biological resource in various industries, such as food, pharmaceuticals, textiles, paper, pulp, and energy [6,7]. Owing to its high industrial value, cellulase accounts for approximately 20% of the global enzyme market and is expected to create high economic added value as a key resource for future eco-friendly agriculture and alternative energy production [8–10]. Cellulase has been isolated from various microorganisms including *Bacillus* [11], *Clostridium* [12], *Streptomyces* [13], and *Aspergillus* [14]. Therefore, the discovery of microbial resources, which are diversely distributed in nature, can contribute to securing cellulase resources that can be commercialized in the future [15–18].

In this study, in order to secure new cellulase resources, we isolated a cellulase-producing bacterial strain from the soil and identified it as *Paenibacillus peoriae* MK1. Cellulase from this strain was cloned, expressed in *Escherichia coli*, and purified. The purified cellulase was functionally characterized by investigating the effects of metal ions, pH, temperature, and substrate specificity and reaction conditions were optimized for the hydrolysis of cellulose. Cellulase from *P. peoriae* MK1 exhibited the highest activity toward CM-cellulose among previously reported cellulases and showed the highest stability among the reported cellulases from *Paenibacillus* spp.

## 2. Materials and Methods

### 2.1. Materials

Carboxymethyl cellulose (CM-cellulose) was purchased from JUNSEI (Tokyo, Japan) and sigmacell cellulose and α-cellulose were purchased from Sigma (St. Louis, MO, USA). To prepare swollen cellulose, 5 g of Avicel PH-101 (Sigma, St. Louis, MO, USA) was added to 200 mL of 35% NaOH, mixed for 30 min, and then 3 L of ice water was added. After that, the pH was neutralized to 7.0 using HCl and at 4 °C. The supernatant was removed from the completely precipitated Avicel and this process was repeated 5 times. The sample obtained by adding 500 mL of distilled water was used as 1% swollen cellulose. Unless otherwise mentioned, all other chemicals containing metal ions were purchased from Duksan (Ansan, Republic of Korea).

### 2.2. Isolation and Identification of a Cellulase-Producing Strain

A soil sample collected from soil surrounding naturally decomposed wood in Mungyeong, Korea, was used to screen the cellulase-producing strain. The collected sample was serially diluted 10 fold in distilled water to concentrations (mg/mL) of $10^{-1}$, $10^{-2}$, $10^{-3}$, and $10^{-4}$. The diluted samples were plated on Luria-Burtani (Difco, Sparks, MD, USA) agar plates containing carboxymethyl cellulose (CM-cellulose; JUNSEI, Tokyo, Japan) and trypan blue (Sigma, St. Louis, MO, USA) and then incubated at 37 °C for 24 h. The strains that formed active zones were streaked on the same medium, followed by a secondary culture at 37 °C for 24 h, and then a single colony that formed a clear active zone was isolated as a cellulase-producing strain. Then, 16S rRNA sequencing was performed to identify the isolated strain at the Macrogen facility (Deajeon, Republic of Korea).

### 2.3. Bacterial Strain, Plsmids, and Cloning

*P. peoriae* MK1, *E. coli* BL21 (DE3) (Thermo Fisher Scientific, Waltham, MA, USA) and pET-28a (+) (Novagen, Darmstadt, Germany) were used as the sources of cellulase genomic DNA, host cells, and expression vectors, respectively. The genomic DNA from *P. peoriae* MK1 was extracted using a Exgene™ Cell SV mini kit (GeneAll, Seoul, Republic of Korea). The gene (1722 bp) encoding the putative cellulase was amplified by PCR using *P. peoriae* MK1 genomic DNA as a template. The oligonucleotide primer sequences used for gene

cloning were based on the cellulase DNA sequence of *P. peoriae* (GenBank accession number WP_010346842.1). The forward primer (5′-AA<u>CATATG</u>GAATCTGACGGACAAGCACCAC-3′) and reverse primer (5′-TT<u>CTCGAG</u>TTAGGATGTCGTTCCCGTTACA-3′) were designed to introduce the underlined NdeI and XhoI restriction sites and were synthesized by Bioneer (Daejeon, Republic of Korea). The DNA fragment amplified by PCR using Ex Taq polymerase (TaKaRa, Shiga, Japan) was extracted using a gel extraction kit (ELPIS, Daejeon, Republic of Korea) and cloned into a pLPS T-vector (ELPIS, Daejeon, Republic of Korea). The NdeI-XhoI fragment from the T-vector harboring the gene encoding cellulase was subcloned into the same sites of pET-28a (+), transformed into *E. coli* BL21 (DE3), and plated on Luria-Bertani (LB) agar containing 40 μg/mL of kanamycin. And then, plasmid DNA from a kanamycin-resistant colony was isolated with an Exprep™ Plasmid SV kit (GeneAll, Daejeon, Republic of Korea). DNA sequencing was conducted at Macrogen.

### 2.4. Culture Conditions for Enzyme Expression and Enzyme Purification

Recombinant *E. coli* cells expressing cellulase from *P. peoriae* MK1 were cultivated in 400 mL of LB medium in a 2000-mL flask containing 40 μg/mL of kanamycin at 37 °C with shaking at 250 rpm. When the optical density of the *E. coli* cells reached 0.6 at 600 nm, 0.1 mM isopropyl-β-D-thiogalactopyranoside (IPTG) was added to induce cellulase expression and the culture was incubated at 16 °C for 16 h with shaking at 100 rpm. The cultured cells were harvested from the broth by centrifugation at $6000 \times g$ for 30 min at 4 °C, washed twice with 25 mM 3-[4-(2-hydroxyethyl)-1-piperazinyl]-propane sulfonic acid (EPPS) buffer (pH 7.5), and then resuspended in 50 mM potassium phosphate buffer (pH 8.0) containing 300 mM NaCl, 10 mM imidazole, and 0.1 mM phenylmethylsulfonyl fluoride (PMSF) as a protease inhibitor. Resuspended cells were disrupted on ice using an ultrasonicator (VCX 130; SONICS, Newtown, CT, USA). Cell debris was removed by centrifugation at $10,000 \times g$ for 30 min at 4 °C and the supernatant was filtered through a 0.45-μm filter. The filtrate was applied to a Mini Profinity IMAC Cartridge column (Bio-Rad, Hercules, CA, USA) equilibrated with potassium phosphate buffer (pH 8.0). The column was washed extensively with the same buffer and the bound protein was eluted using a linear gradient of 10–250 mM imidazole at a flow rate of 1 mL/min. Active fractions were collected and dialyzed against 50 mM citric acid buffer (pH 5.0). The resulting solution was used as the purified enzyme.

### 2.5. Determination of Molecular Mass

The expression and purification levels of cellulase were examined by SDS-PAGE using a pre-stained protein ladder (MBI Fermentas, Glen Burnie, MD, USA) as a molecular mass reference. All protein bands were visualized using Coomassie brilliant blue. The total molecular mass of native cellulase from *P. peoriae* MK1 was determined by gel-filtration chromatography using a Sephacryl® S-300 HR column. The purified enzyme was applied to the column, eluted with 50 mM potassium phosphate buffer (pH 7.0) containing 150 mM NaCl at a flow rate of 0.5 mL/min, and detected at a wavelength of 280 nm using a fast protein liquid chromatography system (Bio-Rad, Hercules, CA, USA). The column was calibrated with well-defined protein standards (GE Healthcare, Piscataway, NJ, USA), such as ribonuclease A, ovalbumin, conalbumin, aldolase, and ferritin, with molecular weights of 13.7, 43, 75, 158, and 440 kDa, respectively. The retention times of the protein standards were compared with that of native cellulase from *P. peoriae* MK1 to determine its molecular mass.

### 2.6. Enzyme Assay

The activity of cellulase from *P. peoriae* MK1 was determined by measuring reducing sugar using the dinitrosalicylic acid (DNS) method [19] after reaction using 140 U/mL enzyme and 10 g/L substrate in 50 mM citric acid buffer (pH 5.0) at 40 °C for 10 min. One unit (U) of cellulase activity was defined as the amount of enzyme required to produce 1 μmol of reducing sugar from substrate per min. The substrate specificity was investigated using substrates such as CM-cellulose, swollen cellulose, sigmacell cellulose, and α-cellulose.

## 2.7. Effects of pH, Temperature, and Metal Ions

To evaluate the effects of pH and temperature on the activity of cellulase from *P. peoriae* MK1 toward CM-cellulose, the pH was varied from 3.0 to 9.0 using 50 mM citric acid buffer (pH 3.0–6.0), 50 mM potassium phosphate buffer (pH 6.0–8.0), and 50 mM Tris-HCl buffer (pH 8.0–9.0) and the temperature was varied from 30 °C to 70 °C. The effect of temperature on the stability of cellulase from *P. peoriae* MK1 was monitored as a function of incubation time by incubating the enzyme solution in 50 mM citric acid buffer (pH 5.0) at different temperatures (30, 35, 40, 45, 50, and 55 °C). After incubation, the enzyme solutions were assayed in 50 mM citric acid buffer (pH 5.0) at 40 °C for 10 min. The experimental data for enzyme inactivation were fitted to a first order curve. The rate constant ($k_d$, $min^{-1}$) was determined from the slope of the deactivation time course according to Equation (1). $E_t$ and $E_0$ are the residual enzyme activity after heat treatment for time (t) and the initial enzyme activity before heat treatment, respectively. The half-life of thermal deactivation ($t_{1/2}$) was calculated using Equation (2).

$$\ln(E_t/E_0) = -k_d t, \tag{1}$$

$$t_{1/2} = \ln(2)/k_d, \tag{2}$$

To measure the dependence of cellulase from *P. peoriae* MK1 on metal ions, the purified enzyme was treated with 20 mM ethylene diamine tetra acetic acid (EDTA) at 37 °C for 2 h and dialyzed against 50 mM citric acid buffer (pH 5.0) to prepare an EDTA-treated enzyme. The reactions were performed using EDTA-treated enzymes with 1 mM concentrations of various metal ions, such as $Co^{2+}$ ($CoCl_2 \cdot 6H_2O$), $Fe^{3+}$ ($FeCl_3 \cdot 6H_2O$), $Cu^{2+}$ ($CuSO_4 \cdot 4H_2O$), $Mn^{2+}$ ($MnSO_4 \cdot 7H_2O$), $Mg^{2+}$ ($MgSO_4 \cdot 7H_2O$), $Ni^{2+}$ ($NiCl_2 \cdot 6H_2O$), and $Ba^{2+}$ ($Ba(OH)_2 \cdot 8H_2O$) in 50 mM citric acid buffer (pH 5.0) at 40 °C for 10 min.

## 2.8. Optimization of Enzyme and Substrate Concentrations

The optimal enzyme concentration for the hydrolysis of CM-cellulose was determined by varying the concentration of cellulase from *P. peoriae* MK1 from 81 to 155 U/mL with 10 g/L CM-cellulose and 1 mM $Ca^{2+}$ in 50 mM citric acid buffer (pH 5.0) at 40 °C for 10 min. The optimal substrate concentration for the hydrolysis of CM-cellulose was determined by varying the concentration of CM-cellulose from 2.5 to 30 g/L with 133 U/mL cellulase from *P. peoriae* MK1 and 1 mM $Ca^{2+}$ in 50 mM citric acid buffer (pH 5.0) at 40 °C for 10 min. The time-course reactions for hydrolysis of CM-cellulose by cellulase from *P. peoriae* MK1 were investigated in 50 mM citric acid buffer (pH 5.0) containing 133 U/mL cellulase from *P. peoriae* MK1 and 20 g/L CM-cellulose with 1 mM $Ca^{2+}$ at 40 °C for 1 h.

## 2.9. Analytical Methods

The degradation of CM-cellulose by cellulase from *P. peoriae* MK1 was identified using thin-layer chromatography (TLC). Overall, 10 microliters of the reaction solution and 10 g/L glucose (DUKSAN, Ansan, Republic of Korea), cellobiose (Alfa Aesar, Haverhill, MA, USA), and cellotriose (ChemCruz, Dallas, TX, USA) were spotted on a TLC silica gel 60 plate (Merck Darmstadt, Germany). The plate was developed using n-butanol/ethanol/chloroform/25% ammonia solution (4:5:2:8 by volume). The spots were detected by spraying the plate with a vanillin–sulfuric acid mixture, followed by holding at 200 °C for 10 min.

## 3. Results and Discussion

### 3.1. Cloning, Expression, and Purification of Cellulase from Isolated Strain

Cellulase-producing activity was measured using microorganisms isolated from the soil. Among them, the bacterium with the highest activity was selected. Since the full 16S rRNA sequence of the isolated strain showed maximum identity (98.91%) to *P. peoriae*, the isolated bacterium was identified as a strain of *P. peoriae* and named *P. peoriae* MK1. Therefore, the 1719 bp gene encoding a cellulase family glycosylhydrolaze from *P. peoriae*

with the same sequence in GenBank (Accession No., WP_010346842.1) was cloned and expressed in *E. coli* BL21 (DE3) in a soluble form. The amino acid sequence of the expressed enzyme showed 66.7%, 88.1%, and 88.8% identity with the glycoside hydrolase (GH) 5 family glycosidases from *Paenibacillus xylanilyticus* [20], *Paenibacillus polymyxa* [21], and *Paenibacillus* sp. [22], respectively.

Cellulase from *P. peoriae* MK1 was purified using HisTrap affinity chromatography into a soluble protein with a 13.2-fold purification, a final yield of 19%, and a specific activity of 77 μmol/min/mg for CM-cellulose. The molecular mass of the purified enzyme was approximately 65 kDa, as determined by SDS-PAGE (Figure 1a), which was consistent with the calculated value of 64,512 Da based on 574 amino acids combined with six histidine residues. The molecular mass of the cellulase from *P. peoriae* MK1 was similar to that of cellulases from *Paenibacillus* sp., *P. polymyxa*, and *P. xylanilyticus* (Table 1). Gel-filtration chromatography showed that the total molecular mass of the native cellulase from *P. peoriae* MK1 was 65 kDa in comparison to the retention times of the calibrated standard proteins (Figure 1b), indicating that the protein exists as a monomer.

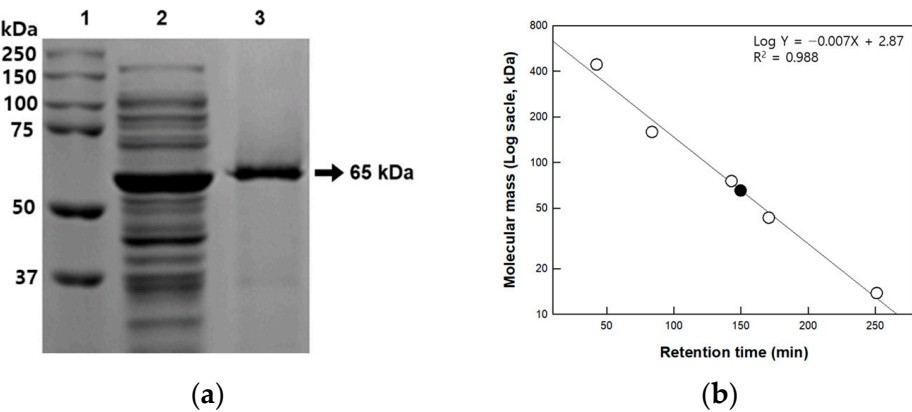

**(a)**　　　　　　　　　**(b)**

**Figure 1.** Molecular mass determination of recombinant cellulase from *P. peoriae* MK1. (**a**) SDS-PAGE analysis. Lane 1, molecular weight marker proteins; lane 2, crude enzyme extract; and lane 3, purified cellulase from *P. peoriae* MK1. (**b**) Calibration curve for gel-filtration. The reference proteins (○) such as ferritin (440 kDa), aldolase (158 kDa), conalbumin (75 kDa), ovalbumin (43 kDa), and ribonuclease A (13.7 kDa) were used for determination of the total molecular mass of cellulase from *P. peoriae* MK1 (●).

**Table 1.** Comparison of characteristics with cellulases belonging to GH family 5.

| Strain | Molecular Mass (kDa) | Metal Ion | pH | Temperature (°C) | Reference |
|---|---|---|---|---|---|
| *B. amyloiquefaciens* | 54 | $Ca^{2+}$ | 7.0 | 50 | [23] |
| *B. subtilis* | 55 | NR * | 8.0 | 50–60 | [24] |
| *B. subtilis* BY-4 | 55 | $Mg^{2+}$ | 4.5 | 60 | [25] |
| *Paenibacillus* sp. | 63.5 | $Co^{2+}$ | 5.0 | 40 | [22] |
| *P. barcinonensis* | 58.6 | $Fe^{2+}$ | 7.0 | 35 | [26] |
| *P. campinasensis* | 38 | NR * | 6.0–7.0 | 60 | [27] |
| *P. peoriae* MK1 | 65 | $Cu^{2+}$ | 5.0 | 40 | This study |
| *P. polymyxa* | 61 | NR * | 6.0 | 50 | [21] |
| *P. terrae* | NR | NR * | 5.5 | 50 | [28] |
| *P. xylanilyticus* | 64 | $Cu^{2+}$ | 6.0 | 40 | [20] |
| *T. halotolerans* | 49.6 | $Ca^{2+}$ | 8.0 | 50 | [29] |

* NR, not reported.

### 3.2. Effect of Metal Ions on the Activity of Cellulase from P. peoriae MK1

The effect of metal ions on the cellulase activity was evaluated at a concentration of 1 mM (Figure 2). $Cu^{2+}$ enhanced the activity of cellulase from *P. peoriae* MK1 by 128%, followed by $Ba^{2+}$, $Mg^{2+}$, and $Fe^{2+}$ by 118, 115, and 111%, respectively, whereas $Mn^{2+}$ and $Ni^{2+}$ inhibited the activity. These results are similar to $Cu^{2+}$ increasing in the relative

activities of cellulases from *Paenibacillus* sp. and *P. xylanilyticus* by 125% and 135% [20,22], respectively [20,22], but are in contrast to the decrease in the relative activity of cellulase from *P. campinasensis* by 18% [27]. In contrast, cellulase from *P. peoriae* MK1 was not significantly affected by EDTA treatment, indicating that this enzyme is metal-independent. Among other GH5 family cellulases, those derived from *Paenibacillus* sp. [22] and *P. campinasensis* [27] are metal-independent, whereas those derived from *P. xylanilyticus* [20], *T. halotolerans* [29], and *P. barcinonensis* [26] are metal-dependent. These results indicate that within the GH5 family, the dependence on metal ions may differ for each enzyme.

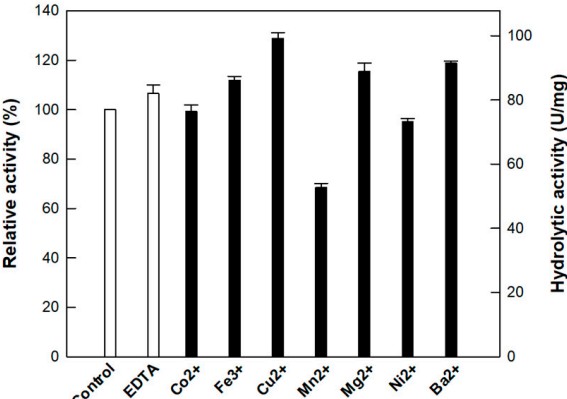

**Figure 2.** Effects of metal ions on the activity of cellulase from *P. peoriae* MK1. Data represent the mean of values from three experiments and error bars represent the standard deviation.

### 3.3. Effects of pH and Temperature on the Activity of Cellulase from P. peoriae MK1

The activity of cellulase from *P. peoriae* MK1 was examined in the pH range 3.0 to 9.0 and the maximal activity was observed at pH 5.0 (Figure 3a). At pH 6.0, the activity reached approximately 77% of its maximum. These results are consistent with most cellulases from *Paenibacillus* spp., such as *Paenibacillus*. sp. [22], *P. xylanilyticus* [20], *P. barcinonensis* [26], and *P. polymyxa* [21] which exhibit optimal activity at pH 5–6 (Table 1). The temperature was varied from 30 to 70 °C to investigate its effect on the activity of cellulase from *P. peoriae* MK1 and the activity was maximal at 40 °C (Figure 3b). Maximal activities of other cellulases from *Paenibacillus* spp. such as *P. xylanilyticus* [20] and *Paenibacillus*. sp. [22]; *P. terrae* [28]; *P. campinasensis* [27]; and *P. barcinonensis* [26] were observed at 40, 50, 60, and 65 °C, respectively (Table 1).

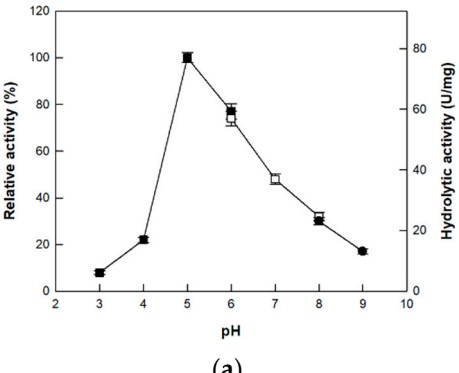 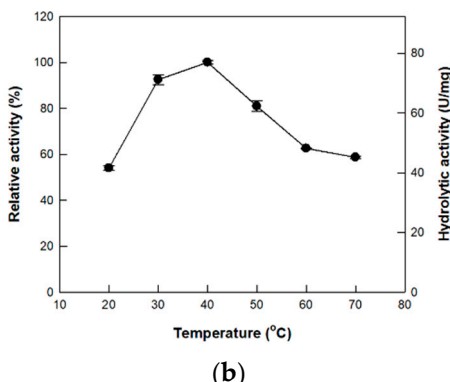

(**a**)                                          (**b**)

**Figure 3.** Effects of pH and temperature on the activity of cellulase from *P. peoriae* MK1. (**a**) Effect of pH. The reactions were performed in 50 mM citric acid buffer (pH 3.0–6.0; ■), potassium phosphate buffer (pH 6.0–8.0; □), and Tris-HCl buffer (pH 8.0–9.0; ●) at 40 °C for 10 min. (**b**) Effect of temperature. The reactions were performed in 50 mM citric acid buffer (pH 5.0) at different temperatures, ranging from 30 °C to 70 °C for 10 min. Data represent the mean of values from three experiments and error bars represent the standard deviation.

The thermal stability of cellulase from *P. peoriae* MK1 was examined in the temperature range of 30–55 °C (Figure 4). The thermal deactivation rate constant $k_d$ (min$^{-1}$) at 40 °C, the temperature at which the enzyme showed maximum activity, was found to be approximately 1.4- and 15-fold slower than the rates at 45 and 50 °C, respectively (Table 2). On the other hand, the rates at 30 and 35 °C were only 0.3- and 0.1-fold slower than the rate at 40 °C, respectively, suggesting that 40 °C is an appropriate temperature for the hydrolysis of cellulose by cellulase from *P. peoriae* MK1. The half-lives of the enzyme were 105.6, 41.0, 13.4, 9.5, and 0.9 h at 30, 35, 40, 45, and 50 °C, respectively; at 55 °C it was not measurable because of the loss of activity after 10 min. Cellulase from *P. peoriae* MK1 showed higher thermostability than cellulase from *P. xylanilyticus* [20], which displayed residual activity of 86% for 1 h at 40 °C.

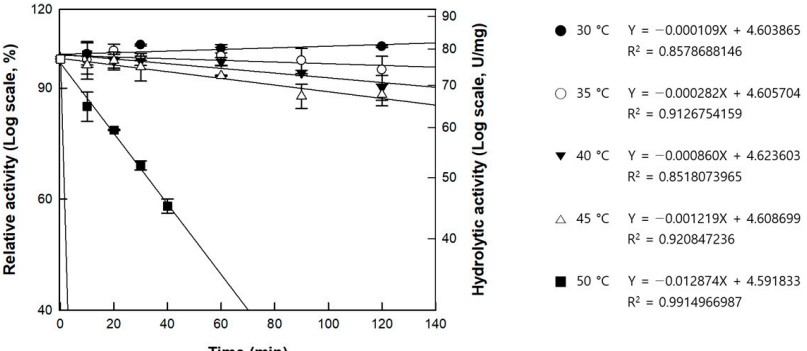

**Figure 4.** Thermal inactivation of cellulase from *P. peoriae* MK1. The enzyme was incubated at 30, 35, 40, 45, 50, and 55 °C, withdrawn at each time point, and assayed in 50 mM citric acid buffer (pH 5.0) at 40 °C for 10 min. Data represent the mean of values from three experiments and error bars represent the standard deviation.

**Table 2.** Deactivation constant ($k_d$) and half-lives ($t_{1/2}$) of cellulase from *P. peoriae* MK1 at 30, 35, 40, 45, and 50 °C.

| Temperature (°C) | $k_d$ (min$^{-1}$) | $t_{1/2}$ (h) |
|---|---|---|
| 30 | $1.09 \times 10^{-4}$ | 105.6 |
| 35 | $2.82 \times 10^{-4}$ | 41.0 |
| 40 | $8.60 \times 10^{-4}$ | 13.4 |
| 45 | $1.22 \times 10^{-3}$ | 9.5 |
| 50 | $1.29 \times 10^{-2}$ | 0.9 |

### 3.4. Substrate Specificity of Cellulase from P. peoriae MK1

The substrate specificity of cellulase from *P. peoriae* MK1 was investigated using amorphous substrates such as CM-cellulose and swollen cellulose and crystalline substrates such as sigmacell cellulose and α-cellulose (Table 3). The enzyme activity was the highest for CM-cellulose (77.0 U/mg), followed by swollen cellulose, sigmacell cellulose, and α-cellulose. Cellulase from *P. peoriae* MK1 showed 1.9-fold higher activity toward CM-cellulose than cellulase from *P. barcinonensis*, which has the highest activity toward CM-cellulose among previously reported cellulases [26].

**Table 3.** Substrate specificity of cellulase from *P. peoriae* MK1.

| Substrate | Enzyme Unit (U/mg) |
|---|---|
| CM-cellulose | $77.0 \pm 0.03$ |
| Swollen cellulose | $15.4 \pm 0.01$ |
| Sigmacell cellulose | $10.6 \pm 0.03$ |
| α-Cellulose | $6.3 \pm 0.02$ |

The enzyme activity toward swollen cellulose, whose hydrogen bonds are optionally removed by NaOH, was lower than that for CM-cellulose but 1.5- and 2.4-fold higher than those for crystalline celluloses such as sigmacell cellulose and α-cellulose, respectively. This was similar to the finding that cellulase from *P. campinasensis* showed higher activity toward amorphous CM-cellulose than that toward crystalline cellulose [27].

*3.5. Hydrolysis of CM-Cellulose by Cellulase from P. peoriae MK1 on the Optimized Enzyme and Substrate Concentrations*

Hydrolysis of CM-cellulose was optimized by varying the enzyme and substrate concentrations (Figure 5). The concentration of reducing sugar produced increased with increasing enzyme concentration up to 133 U/mL; however, above this concentration, the production of reducing sugar plateaued, indicating that the optimum enzyme concentration was 133 U/mL. As the concentration of CM-cellulose as a substrate increased, the production of reducing sugars increased proportionally up to 20 g/L of substrate. However, the production rate decreased above 20 g/L of CM-cellulose and this substrate concentration was used as the optimized concentration.

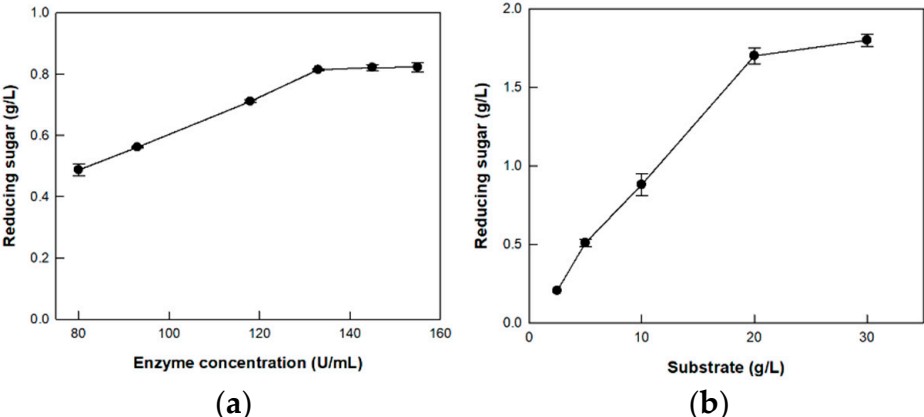

**(a)**　　　　　　　　　　　　　　**(b)**

**Figure 5.** Effects of concentrations of enzyme and substrate on the hydrolysis of CM-cellulose. (**a**) Effect of the concentration of cellulase from *P. peoriae* MK1. The reactions were performed in 50 mM citric acid buffer (pH 5.0) containing 10 g/L CM-cellulose at 40 °C for 10 min with varying concentrations of enzyme from 81 to 139 U/mL. (**b**) Effect of concentration of CM-cellulose. The reactions were performed in 50 mM citric acid buffer (pH 5.0) containing 135 U/mL enzyme at 40 °C for 10 min with varying concentrations of substrate from 2.5 to 30 g/L. Data represent the mean of values from three experiments and error bars represent the standard deviation.

The pH, temperature, and metal ion at which cellulase from *P. peoriae* MK1 showed the highest activity were 5.0, 40 °C, and $Ca^{2+}$, respectively, and the optimal enzyme and substrate concentrations were 133 U/mL and 20 g/L of CM-cellulose, respectively. Under these conditions, a time-course reaction for the hydrolysis of CM-cellulose was performed and analyzed both qualitatively and quantitatively. During the entire reaction time, most of the products were oligosaccharides other than glucose, cellobiose, and cellotriose (Figure 6a), indicating that the cellulase from *P. peoriae* MK1 is an endo-type cellulase that produces oligosaccharides through random hydrolysis of substrates. The production of reducing sugars increased over time and reached saturation at 40 min. However, the production rate rapidly decreased after 10 min (Figure 6b). Therefore, the optimal reaction time was determined to be 10 min and the cellulase from *P. peoriae* MK1 showed a productivity of 11.1 g/L/h for 10 min.

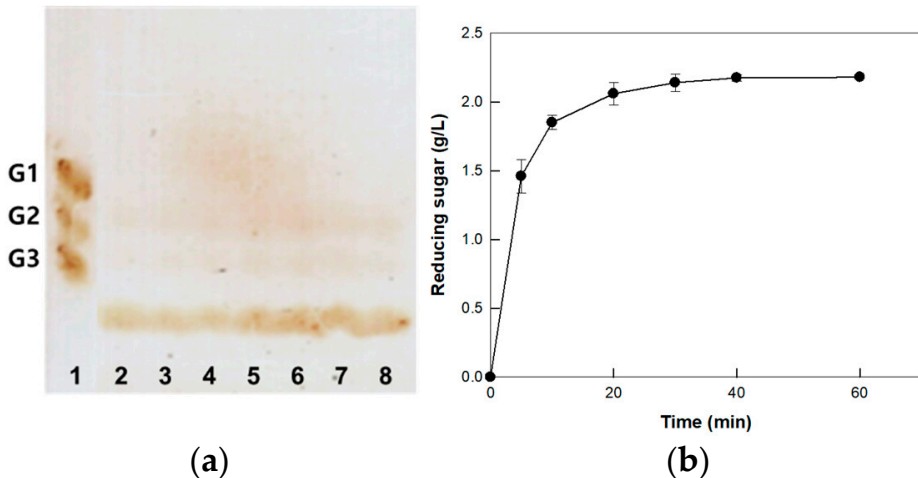

**Figure 6.** (**a**) Qualitative and (**b**) quantitative analysis of time course reactions for the hydrolysis of CM-cellulose. The reactions were performed in 50 mM citric acid buffer (pH 5.0) containing 135 U/mL enzyme and 20 g/L CM-cellulose at 40 °C for 60 min. Data represent the mean of values from three experiments and error bars represent the standard deviation. Lanes 2–8 of TLC represent hydrolysis products for 5, 10, 20, 30, 40, 60 min, and overnight, respectively. Lane 1 contains standards of glucose (G1), cellobiose (G2), and cellotriose (G3).

## 4. Conclusions

A cellulase-producing bacterial strain was isolated from the soil and identified as *P. peoriae* MK1 and the cellulase from the strain was cloned, expressed, purified, and characterized. The cellulase from *P. peoriae* MK1 was identified as a monomer- and metal-independent enzyme. The activity of the enzyme was enhanced most efficiently by $Cu^{2+}$ among metal ions and the highest activity was observed at pH 5.0 and 40 °C. Cellulase from *P. peoriae* MK1 exhibited the highest stability at the optimum temperature among the reported cellulases from *Paenibacillus* spp. and the highest activity toward CM-cellulose as an amorphous substrate. Under optimized concentrations of the enzyme (133 U/mL) and substrate (20 g/L), the cellulase from *P. peoriae* MK1 is an endo-type cellulase and hydrolyzes CM-cellulose to reducing sugars; it is composed mostly of oligosaccharides with a productivity of 11.1 g/L/h for 10 min. These findings will be helpful for securing cellulase resources with high industrial usability.

**Author Contributions:** Conceptualization, K.-C.S., Y.-S.K. and C.-S.P.; Methodology, K.-C.S. and S.J.K.; Validation, K.-C.S., D.W.K. and S.J.K.; Investigation, S.J.K., K.-C.S., Y.-S.K. and C.-S.P.; Writing, K.-C.S., Y.-S.K. and C.-S.P.; Supervision, Y.-S.K. and C.-S.P.; Funding Acquisition, C.-S.P. and Y.-S.K. All authors have read and agreed to the published version of the manuscript.

**Funding:** This work was supported by National Research Foundation of Korea (NRF) grants funded by the Korean government (MSIT) (grant numbers 2022R1F1A1065362 and 2018R1D1A1B07050820).

**Institutional Review Board Statement:** Not applicable.

**Informed Consent Statement:** Not applicable.

**Data Availability Statement:** Not applicable.

**Acknowledgments:** This study was supported by the KU Research Professor Program at Konkuk University.

**Conflicts of Interest:** The authors declare no conflict of interest.

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
