# Peer review of "Cloning and Characterization of Cellulase from Paenibacillus peoriae MK1 Isolated from Soil"

_fermentation, doi:10.3390/fermentation9100873_

Round 1
Reviewer 1 Report
Dear authors, I think your paper could be improved by adressing the following issues:
1.Linees 120-122 mention the way the calibration of a gel chromatography system. How was the detection performed and what were the conditions of operating this system. The calibration curve was not shown....please add it as a supplementary material.
2. Line 130 " reducing sugars"...please specify the analytical method used for the determination.
3. The section 2.7 " Optimization of reaction needs a clear expression of the optimization criteria. In fact, in my personal opinion a DOE approach could have been appropriate.
4. Figure 1 b) .... please show the correlation equation.
5. Figure 3 b)... what happens at temperatures below 30 Celsius degrees?
I did not attach any file because I do not consider it necessary.
There are some minor spelling or typing mistakes which from my opinion need to be changed. Ex. Plsmids line 78.
Author Response
Q1. Lines 120-122 mention the way the calibration of a gel chromatography system. How was the detection performed and what were the conditions of operating this system. The calibration curve was not shown. Please add it as a supplementary material.
Answer) Thank you for your comment. As you suggested, we added the conditions of operation and detection in the revised manuscript as followed: “The purified enzyme was applied to the column, eluted with 50 mM phosphate buffer (pH 7.0) containing 150 mM NaCl at a flow rate of 0.5 mL/min, and detected at a wavelength of 280 nm using a fast protein liquid chromatography system (Bio-Rad, Hercules, CA, USA).”. (Line 134−137 of the revised manuscript) Additionally, the calibration curve was already shown in Figure 1b of the original manuscript. For better understanding, we changed “Gel-filtration chromatography” to “Calibration curve for gel-filtration” in the revised manuscript. (Line 214 of the revised manuscript)
Q2. Line 130 "reducing sugars"...please specify the analytical method used for the determination.
Answer) Thank you for your concern. We had already mentioned “The activity of cellulase from P. peoriae MK1 was measured by the dinitrosalicylic acid (DNS) method”. (Line 127−128 of the original manuscript) For better understanding, we modified the relevant sentence as follows: “The activity of cellulase from P. peoriae MK1 was determined by measuring reducing sugar using the dinitrosalicylic acid (DNS) method” (Line 144−145 of the revised manuscript)
Q3. The section 2.7 " Optimization of reaction needs a clear expression of the optimization criteria. In fact, in my personal opinion a DOE approach could have been appropriate.
Answer) Thank you for your suggestion. In this study, we only confirmed the optimal substrate and enzyme concentrations rather than optimizing the conditions for the entire reaction. Therefore, to reduce misunderstanding, “reaction conditions” were changed to “enzyme and substrate concentrations” and related sentences were modified in the revised manuscript. (Line 21, 172, 283−284, and 301−302 of the revised manuscript)
Q4. Figure 1 b) .... please show the correlation equation.
Answer) Thank you for your suggestion. As you suggested, we newly added the correlation equation in Figure 1b of the revised manuscript.
Q5. Figure 3 b)... what happens at temperatures below 30 Celsius degrees?
Answer) Thank you for your question. We newly conducted a test on the activity of cellulase from P. peoriae MK1 at 20 °C and confirmed that it showed a relative activity of 54%. Therefore, Figure 3 b was modified to include this result in the revised manuscript.
Reviewer 2 Report
In this study, a recombinant cellulase from P. peoriae MK1 was produced using E. coli as a host. The enzyme was purified and their properties were characterized. This is an interesting study that deserves publication in this jornal. However, some questions should be addressed, as suggested.
1. Introduction: How is this system different to other reports to merit publication? Please, report.
2. Materials and methods: a) I recommend provide possible suppliers for all chemicals and microrganisms used in this study. (Section 2.1 for Materials); b) Line 70: Put units for concentrations.; c) Line 97: “Expression” instead of “Epression”.; d) Lines 105/111/136: sodium or potassium???? Please, explain.; e) Lines 121-125: Possible suppliers for the protein standards.; f) Lines 137-141: Thermal stability tests: The authors should better explain the kinetic model used to estimate relevant parameters. It seems me that the authors fitted a linear inactivation kinetic model to the experimental data (see Figure 4). Please, comment.; g) Line 146: Provide soluble salts used in this study (nitrate or chloride???).
3. Results: a) Table 1: “Molecular mass” instead of “Molecular weight”. b) Figures 2, 3, and 4. Provide experimental values of hydrolytic activity for “100% relative activity”.; c) Thermal stability tests: I recommend that the discussion is improved, provide thermal inactivation constant (kd); half-life (t1/2) and coefficient correlation (R2) values for all these assays (Add in a Table). This is very important to the readers.
In this study, a recombinant cellulase from P. peoriae MK1 was produced using E. coli as a host. The enzyme was purified and their properties were characterized. This is an interesting study that deserves publication in this jornal. However, some questions should be addressed, as suggested.
1. Introduction: How is this system different to other reports to merit publication? Please, report.
2. Materials and methods: a) I recommend provide possible suppliers for all chemicals and microrganisms used in this study. (Section 2.1 for Materials); b) Line 70: Put units for concentrations.; c) Line 97: “Expression” instead of “Epression”.; d) Lines 105/111/136: sodium or potassium???? Please, explain.; e) Lines 121-125: Possible suppliers for the protein standards.; f) Lines 137-141: Thermal stability tests: The authors should better explain the kinetic model used to estimate relevant parameters. It seems me that the authors fitted a linear inactivation kinetic model to the experimental data (see Figure 4). Please, comment.; g) Line 146: Provide soluble salts used in this study (nitrate or chloride???).
3. Results: a) Table 1: “Molecular mass” instead of “Molecular weight”. b) Figures 2, 3, and 4. Provide experimental values of hydrolytic activity for “100% relative activity”.; c) Thermal stability tests: I recommend that the discussion is improved, provide thermal inactivation constant (kd); half-life (t1/2) and coefficient correlation (R2) values for all these assays (Add in a Table). This is very important to the readers.
Author Response
Q1. Introduction: How is this system different to other reports to merit publication? Please, report.
Answer) Thank you for your suggestion. As we already mentioned in the introduction, it is very important to secure cellulase resources with various industrial uses, and for this reason, we discovered new cellulase from P. peoriae MK1 and characterized it. Cellulase from P. peoriae MK1 exhibited the highest activity toward CM-cellulose among previously reported cellulases and showed the highest stability among the reported cellulases from Paenibacillus spp.. Therefore, we have newly added relevant content to the introduction as follows: “in order to secure new cellulase resource” (Line 61 of the revised manuscript) “Cellulase from P. peoriae MK1 exhibited the highest activity toward CM-cellulose among previously reported cellulases and showed the highest stability among the re-ported cellulases from Paenibacillus spp..” (Line 66−68 of the revised manuscript)
Q2. Materials and methods: a) I recommend provide possible suppliers for all chemicals and microrganisms used in this study. (Section 2.1 for Materials); b) Line 70: Put units for concentrations.; c) Line 97: “Expression” instead of “Epression”.; d) Lines 105/111/136: sodium or potassium???? Please, explain.; e) Lines 121-125: Possible suppliers for the protein standards.; f) Lines 137-141: Thermal stability tests: The authors should better explain the kinetic model used to estimate relevant parameters. It seems me that the authors fitted a linear inactivation kinetic model to the experimental data (see Figure 4). Please, comment.; g) Line 146: Provide soluble salts used in this study (nitrate or chloride???).
Answer) Thank you for your comment.
- a) We newly added Section 2.1 for Materials in the revised manuscript. (Line 71−79 of the revised manuscript)
- b) “mg/mL" was added as the unit for concentration in the revised manuscript. (Line 83 of the revised manuscript)
- c) “Epression” was corrected to “Expression” in the revised manuscript. (Line 110 of the revised manuscript)
- d) We used potassium phosphate buffer throughout the entire experiment. Therefore, “phosphate buffer” was corrected to “potassium phosphate buffer” in the revised manuscript. (Line 118, 124, 135, 154, and 247−248 of the revised manuscript)
- e) We had already mentioned the supplier “GE Healthcare” in the original manuscript. To express it clearly, we also indicated the supplier's location as follows: “GE Healthcare, Piscataway, NJ, USA” in the revised manuscript. (Line 138 of the revised manuscript)
- f) For accurate estimation, the experimental data for enzyme inactivation were refitted to a first order curve in the revised manuscript. Therefore, the following sentences have been newly inserted: “The experimental data for enzyme inactivation were fitted to a first order curve. The rate constant (kd, min−1) was determined from the slope of the deactivation time course according to Equation (1). Et and E0 are the residual enzyme activity after heat treatment for time (t) and the initial enzyme activity before heat treatment, respectively. The half-life of thermal deactivation (t1/2) was calculated using Equation (2). ln(Et/E0) = −kdt, (1) t1/2 = ln(2)/kd, (2)”
(Line 159−164 of the revised manuscript)
- g) For specific purposes, soluble salts of metal ions were presented in the revised manuscript. (Line 169−171 of the revised manuscript)
Q3. Results: a) Table 1: “Molecular mass” instead of “Molecular weight”. b) Figures 2, 3, and 4. Provide experimental values of hydrolytic activity for “100% relative activity”.; c) Thermal stability tests: I recommend that the discussion is improved, provide thermal inactivation constant (kd); half-life (t1/2) and coefficient correlation (R2) values for all these assays (Add in a Table). This is very important to the readers.
Answer) Thank you for your comment.
- a) “Molecular weight” was corrected to “Molecular mass” in Table 1 of the revised manuscript.
- b) As you suggested, we added “Hydrolytic activity (U/mg)” in right Y axis of Figures 2, 3, and 4 in the revised manuscript.
- c) We retested some data points for thermal stability to minimize errors. As you recommended, the R2 value was shown in Figure 4, and kd and t1/2 were shown in Table 2 of the revised manuscript. In addition, related content was described in “Results and Discussion” as follows: “The thermal deactivation rate constant kd (min−1) at 40 °C, where the enzyme showed maximum activity, was found approximately 1.4- and 15-fold slower than the rates at 45 and 50 °C, respectively (Table 2). On the other hand, the rates at 30 and 35 °C were only 0.3- and 0.1-fold slower than the rate at 40 °C, respectively, suggesting that 40 °C is an appropriate temperature for the hydrolysis of cellulose by cellulase from peoriae MK1.” (Line 253−258 of the revised manuscript)
Round 2
Reviewer 1 Report
Dear Authors, your explanations seem ok.
Reviewer 2 Report
The authors corrected the manuscript, as suggested. Therefore, I recommend its acceptance in the present version.
The authors corrected the manuscript, as suggested. Therefore, I recommend its acceptance in the present version.